# Small-cell carcinoma in the head and neck region: A propensity score-matched analysis of the effect of surgery

**Kiyohito Hosokawa**[ID]**, Yukinori Takenaka**[ID]*****, Takashi Sato, Takeshi Tsuda,
Hirotaka Eguchi, Masami Suzuki, Koji Kitamura, Takahito Fukusumi, Motoyuki Suzuki,
Hidenori Inohara**

Department of Otorhinolaryngology-Head and Neck Surgery, Osaka University Graduate School of Medicine,
Suita, Osaka, Japan

* ytakenaka@ent.med.osaka-u.ac.jp

org/10.1371/journal.pone.0312455

Guangzhou Medical University, CHINA

**Data Availability Statement:** All relevant data are
within the manuscript and its Supporting
Information files.

## Abstract

### Background

Head and neck small-cell carcinoma (HNSmCC) is a rare and aggressive cancer with a high
tendency for distant metastasis. It is treated with multimodal treatment involving chemother-
apy. Occasionally, surgery is performed for the management of locoregional HNSmCC.
However, the benefits of surgery in this context have not yet been elucidated. Therefore, in
this study, we aimed to investigate whether surgery could improve the survival of patients
with HNSmCC.

### Patients and methods

We obtained data from patients with locoregional HNSmCC treated with chemoradiation
therapy (CRT) from the Surveillance, Epidemiology, and End Results database. Patients
who did and did not undergo surgery were matched using propensity scores. The overall
survival (OS) and disease-specific survival (DSS) rates were estimated using the Kaplan-
Meier method and tested using the log-rank test. Hazard ratios (HRs) were calculated using
the Cox proportional hazard model.

### Results

The 5-year OS rates of the patients who did and did not undergo surgery were 57.2% and
50.6%, respectively (P = 0.689); the corresponding 5-year DSS rates were 61.0% and
57.5% (P = 0.769). The adjusted HRs for surgery were 0.85 (95% confidence interval [CI]:
0.54–1.33) for OS and 0.87 (95% CI: 0.51–1.49) for DSS.

### Conclusion

The addition of surgery to CRT did not improve the survival of patients with locoregional
HNSmCC.

**Funding:** This work was supported by a grant awarded to KK by the Japan Society for the Promotion of Science (Grant Number: JP24K12647, https://www.jsps.go.jp/). The funder had no role in study design, data collection, analysis, decision to publish, or preparation of the manuscript. There was no additional external funding received for this study.

**Competing interests:** The authors have declared that no competing interests exist.

## Introduction

Head and neck cancer is the sixth most common cancer worldwide [1]. Various histological types of cancers can arise in the head and neck region. Small-cell carcinoma (SmCC) is an aggressive malignancy with a high incidence of distant metastasis. It can develop at any site in the body but arises mostly in the lungs. The common sites for extrapulmonary SmCC are the genitourinary and gastrointestinal organs, and head and neck SmCCs (HNSmCCs) are extremely rare [2–4].

Previous population-based studies have demonstrated the demographics, clinicopathological characteristics, and prognoses of patients with HNSmCC [5, 6]. HNSmCC is diagnosed usually in the fifth decade of life or later, mostly among white men. The salivary gland is the most common site of HNSmCC development, followed by the laryngopharynx and the oral cavity. Distant metastasis has been found in one-third of the patients at diagnosis. The median survival time of the affected patients is 17 months, and the 5- and 10-year overall survival (OS) rates have been reported to be 26% and 18%, respectively [5]. In the Cox proportional hazards model, T classification and radiotherapy were identified as prognostic factors for OS, and stage, N and M stages, and chemotherapy were identified as prognostic factors for disease-specific survival (DSS) [6].

Because of its poor prognosis, multimodal treatment is usually selected for SmCC development at any site. However, evidence for the treatment of extrapulmonary SmCC, including HNSmCC, is lacking. Therefore, multimodal treatment for HNSmCC is based on that for SmCC of the lungs. Combination chemotherapy with a platinum-based anticancer drug and a topoisomerase inhibitor is the main pillar of SmCC therapy [7]. Additionally, radiotherapy plays a role as a definitive and palliative therapy for non-metastatic SmCC. However, the role of surgery in the treatment of SmCC remains controversial.

Surgery plays a major role in the treatment of head and neck malignancies of various histological types. It is occasionally performed for the management of locoregional HNSmCC. However, evidence supporting the use of surgery for HNSmCC is lacking. Hence, this study was aimed at investigating the prognosis of HNSmCC according to the treatment modality and whether surgery could improve the survival of patients with locoregional HNSmCC.

## Patients and methods

The requirement of approval from the Institutional Review Board of Osaka University Hospital was waived because the analyzed data were publicly available and anonymized.

### Data retrieval

Individual patient data were retrieved from the Surveillance, Epidemiology, and End Results (SEER) Research Plus Data, 17 registries, and the November 2020 Sub by using the SEER Stat software version 8.4.3 (National Cancer Institute, Bethesda, MD, USA). The data was accessed on April 15, 2024. The inclusion criteria were as follows: (1) primary site in the head and neck (Site recode ICD-O-3 2023 Revision Expanded, "Head And Neck"), (2) histologically confirmed SmCC (ICD-O-3 Hist/behav, 8041/3, 8042/3, 8043/3, 8044/3, 8045/3), (3) "localized" or "regional" status in the combined summary stage, (4) diagnosis between 2000 and 2020, and (5) treatment with external RT (coded as "beam radiation") and chemotherapy. Cases for which survival data were missing were excluded. SEER does not include personally identifiable information, ensuring patient confidentiality.

## Statistical analysis

Propensity score matching was performed using the nearest-matching method with a caliper of 0.20. The covariates for matching were age, race, sex, tumor extent, and primary site. The associations between categorical variables and between categorical and continuous variables were compared using the chi-square and Kruskal-Wallis tests, respectively. DSS and OS rates were estimated using the Kaplan-Meier method and compared using the log-rank test. Multivariate analysis was performed using a Cox proportional hazards model. Additionally, we employed inverse probability of treatment weighting (IPTW) and performed a weighted Cox proportional hazards model. Statistical significance was set at $P < 0.05$. All statistical analyses were performed using EZR (Saitama Medical Center, Jichi Medical University, Saitama, Japan), which is a graphical user interface for R (R Foundation for Statistical Computing, Vienna, Austria).

## Results

### Patient characteristics

We identified 647 patients with HNSmCC from the SEER database. Among them, we included 229 with locoregional HNSmCC treated with chemoradiation therapy (CRT) (**Fig 1**). The clinicopathological characteristics of the patients are summarized in **Table 1** (**S1 File**). Surgery was performed in 41.9% of the cases. Patients who underwent surgery were older than those who did not. Cases in which the primary site was the salivary glands were more likely to be treated surgically, whereas those in which the primary site was the larynx, and the nasopharynx were more likely to be treated without surgery. To minimize confounding factors, propensity score matching was performed between patients who underwent surgery and those who did not. Seventy-one cases each were included in further analyses (**S2 File**). The kernel density plot of the propensity scores demonstrated the effectiveness of the matching process (**Fig 2**). The median follow-up period for the surviving patients was 56 months.

### Prognostic factors for survival

During the follow-up period, 54 patients died of HNSmCC and 24 of other causes. **Table 2** shows the results of the univariate Cox proportional hazards analysis. Male sex was associated with improved survival (hazard ratio [HR]: 0.56, 95% confidence interval [CI]: 0.35–0.89 for OS and HR: 0.41, 95% CI: 0.24–0.71 for DSS). No other factors, including surgery, were associated with OS or DSS.

### Surgery and prognosis

**Fig 3** shows OS and DSS according to the addition of surgery to CRT. The 5-year OS rates were 57.2% and 50.6% for the CRT with surgery and CRT without surgery groups, respectively (P = 0.689). The corresponding 5-year DSS rates were 61.0% and 57.5% (P = 0.769).

To further investigate the effect of surgery on survival, a Cox proportional hazards model was used (**Fig 4**). After adjustment for age, race, sex and extent of disease, the adjusted HRs for surgery were 0.85 (95% CI: 0.54–1.33) and 0.87 (0.51–1.49) for OS and DSS, respectively. The results of subgroup analyses for disease extent are shown in Fig 2. The unadjusted HR for surgery in cases of localized disease were 0.96 (95% CI: 0.41–2.28) and 1.49 (95% CI: 0.44–5.12) for OS and DSS, respectively. The corresponding values for cases of regional disease were 0.88 (95% CI: 0.52–1.49) and 0.83 (95% CI: 0.45–1.51) for OS and DSS, respectively. Therefore, the addition of surgery to CRT did not improve patient survival.

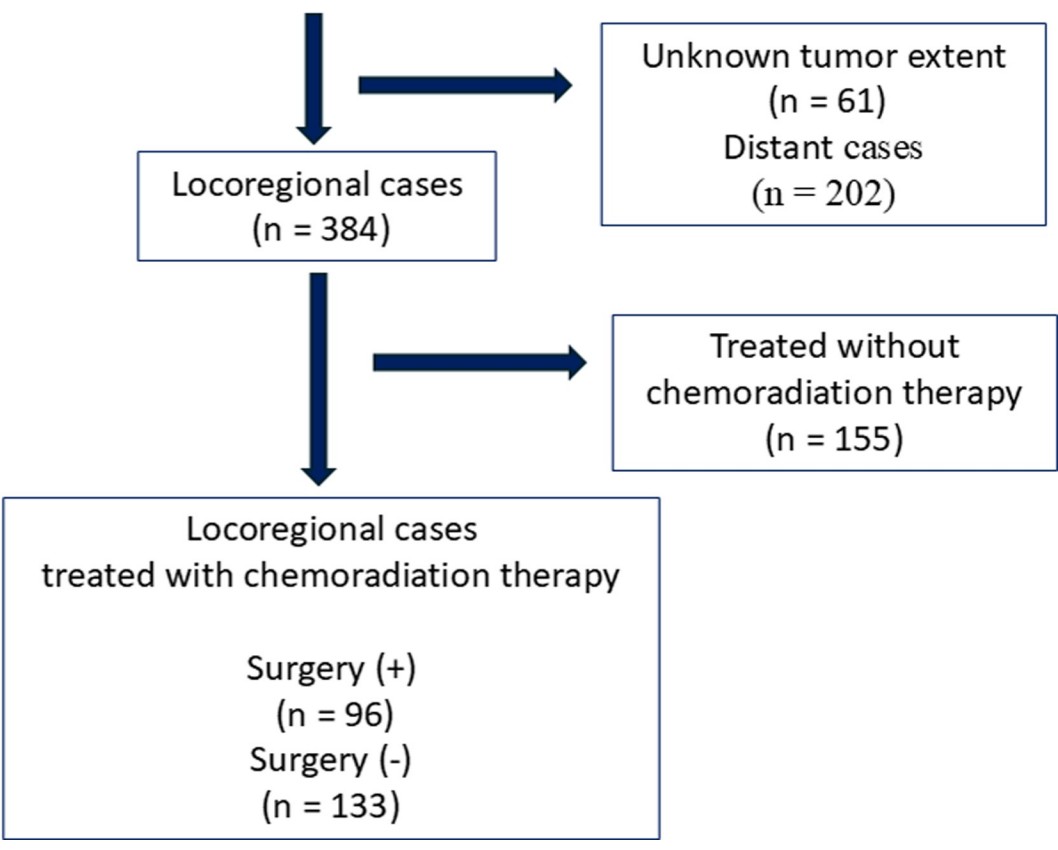

**Fig 1. Flow diagram for the patient selection.**

S1 Fig presents the survival curves for the addition of surgery to CRT, stratified by age, sex, and tumor extent. In all subgroups analyzed, the addition of surgery did not provide a significant survival advantage.

As a sensitivity analysis, we employed IPTW analysis to assess the robustness of our findings. The adjusted HRs for surgery were 0.89 (95% CI: 0.63–1.27) and 0.85 (95% CI: 0.55–1.32) for OS and DSS, respectively.

## Discussion

In this study, we investigated the survival of patients with SmCC who did or did not undergo surgery. After propensity score matching, surgery was found to have no additional benefit for the survival of patients with HNSmCC treated using CRT. Because of the rarity of HNSmCC, conducting a randomized controlled trial is implausible for determining the best treatment strategy. Therefore, our results are the best evidence available thus far.

**Table 1. Patient characteristics.**

| | Before matching | | | | | | After matching | | | | | |
|---|---|---|---|---|---|---|---|---|---|---|---|---|
| | CRT with surgery (n = 96) | | | CRT without surgery (n = 133) | | | | CRT with surgery (n = 71) | | | CRT without surgery (n = 71) | | |
| | Median (IQR) | No. | % | Median (IQR) | No. | % | p value | Median (IQR) | No. | % | Median (IQR) | No. | % | p value |
| Sex | | | | | | | 0.968 | | | | | | | 0.723 |
| Male | | 64 | 66.7 | | 89 | 66.9 | | | 46 | 64.8 | | 48 | 67.6 | |
| Female | | 32 | 33.3 | | 44 | 33 | | | 25 | 35.2 | | 23 | 32.4 | |
| Age, years | 67 (25–89) | | | 61 (22–88) | | | 0.007 | 64 (25–89) | | | 64 (33–88) | | | 0.633 |
| Race | | | | | | | 0.138 | | | | | | | 0.759 |
| White | | 85 | 88.5 | | 109 | 82.0 | | | 63 | 88.7 | | 60 | 84.5 | |
| Black | | 5 | 5.2 | | 17 | 12.8 | | | 5 | 7.0 | | 7 | 9.9 | |
| Other | | 6 | 6.3 | | 7 | 5.3 | | | 3 | 4.2 | | 4 | 5.6 | |
| Primary site | | | | | | | <0.001 | | | | | | | 0.912 |
| Salivary gland | | 40 | 41.7 | | 24 | 18.0 | | | 17 | 23.9 | | 23 | 32.4 | |
| Larynx | | 13 | 13.5 | | 50 | 37.6 | | | 13 | 18.3 | | 10 | 14.1 | |
| Nose and sinus | | 20 | 20.8 | | 17 | 12.8 | | | 20 | 28.2 | | 15 | 21.1 | |
| Oropharynx | | 14 | 14.6 | | 20 | 15.0 | | | 14 | 19.7 | | 14 | 19.7 | |
| Nasopharynx | | 4 | 4.2 | | 15 | 11.3 | | | 4 | 5.6 | | 4 | 5.6 | |
| Hypopharynx | | 1 | 1.0 | | 2 | 1.5 | | | 1 | 1.4 | | 2 | 2.8 | |
| Oral cavity | | 1 | 1.0 | | 3 | 2.3 | | | 1 | 1.4 | | 1 | 1.4 | |
| Other | | 3 | 3.1 | | 2 | 1.5 | | | 1 | 1.4 | | 2 | 2.8 | |
| Extent of disease | | | | | | | 0.737 | | | | | | | 0.694 |
| Localized | | 22 | 22.9 | | 28 | 21.1 | | | 18 | 25.4 | | 16 | 22.5 | |
| Regional | | 74 | 77.1 | | 105 | 78.9 | | | 53 | 74.6 | | 55 | 77.5 | |

Abbreviation: CRT, chemoradiation therapy, IQR, interquartile range

Surgery, radiation therapy, chemotherapy, and immunotherapy are the four pillars of cancer treatment. Squamous cell carcinoma (SCC) is the most common histological type of head and neck cancer. SmCC has clinical characteristics distinct from those of SCC [5]. Compared with SCC of the head and neck (HNSCC), HNSmCC is more likely to develop in female patients and white men and women. The most prominent difference is observed in the proportion of patients with distant diseases. Distant disease was observed in 10% and 23% of HNSCC and SmCC cases, respectively. This difference was reflected in the treatment strategies. Early-stage HNSCC is treated with surgery or radiation therapy [8]. For locoregionally advanced HNSCC, surgery is followed by CRT or definitive CRT. In contrast, HNSmCC, even in the early stages, is treated with multimodal treatment [5, 6, 9]. Moreover, the main pillar of treatment for SmCC is chemotherapy, because of the high metastatic potential of the disease. Among the patients with HNSmCC in the SEER database analyzed in this study, chemotherapy was administered to >70%. Chen et al. compared SmCC in the lungs and that at extrapulmonary sites [4]. In their cohort, 73% of lung SmCCs were treated with chemotherapy and 69% were treated with chemotherapy alone. In contrast, 56% of extrapulmonary SmCCs were treated with chemotherapy and 51% were treated with chemotherapy alone. Matsuyama et al. conducted a multi-institutional retrospective study on HNSmCC [9]. In their study, 89% of patients received chemotherapy. Chemotherapy is administered to many patients with extrapulmonary SmCC, including those with HNSmCC; however, no relevant guidelines exist. In contrast, surgery is performed in a small percentage of patients. Xu et al. investigated

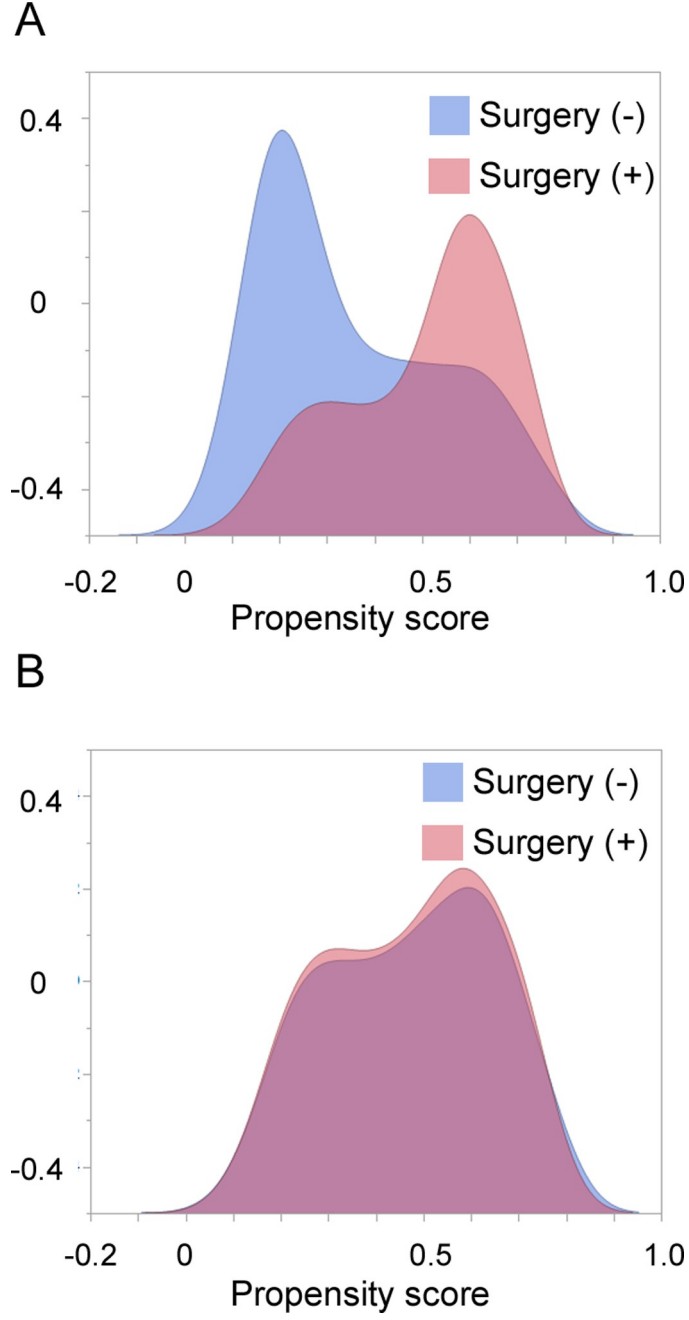

**Fig 2.** Kernel density plot of the propensity scores before matching (A) and after matching (B).

extrapulmonary SmCCs at various sites [2]. The proportion of patients who underwent surgery varied among primary sites: 89% of patients with SmCC in the bladder were treated surgically, whereas only 2% of patients with SmCC in the pancreas were treated surgically. Thus, whether surgery should be incorporated into a multimodal treatment depends on the primary site. Wakeam et al. conducted a propensity score-matched analysis and demonstrated longer survival in patients with stage I/II SmCC treated with surgery followed by chemotherapy than in those treated with CRT [10]. Zhu et al. investigated 458 patients with SmCC in the

**Table 2. Univariate Cox proportional hazards analysis for survival.**

| | Overall survival | | | | | | Disease-specific survival | | | | | |
|---|---|---|---|---|---|---|---|---|---|---|---|---|
| | HR | (95% CI) | | | | P value | HR | (95% CI) | | | | P value |
| Sex | | | | | | | | | | | | |
| Male vs female | 0.56 | ( | 0.35 | – | 0.89 | ) | 0.014 | 0.41 | ( | 0.24 | – | 0.71 | ) | 0.001 |
| Age | | | | | | | | | | | | |
| 65 years or older vs <65 years | 1.49 | ( | 0.95 | – | 2.32 | ) | 0.081 | 1.27 | ( | 0.74 | – | 2.17 | ) | 0.384 |
| Race | | | | | | | | | | | | |
| White | Ref | | | | | | Ref | | | | | |
| Black | 1.08 | ( | 0.47 | – | 2.49 | ) | 0.858 | 1.24 | ( | 0.49 | – | 3.11 | ) | 0.649 |
| Other | 0.51 | ( | 0.16 | – | 1.65 | ) | 0.264 | 0.30 | ( | 0.04 | – | 2.21 | ) | 0.240 |
| Primary site | | | | | | | | | | | | |
| Salivary gland | Ref | | | | | | Ref | | | | | |
| Larynx | 1.80 | ( | 0.94 | – | 3.45 | ) | 0.077 | 1.70 | ( | 0.76 | – | 3.76 | ) | 0.194 |
| Nose and sinus | 0.85 | ( | 0.43 | – | 1.70 | ) | 0.653 | 0.97 | ( | 0.43 | – | 2.19 | ) | 0.939 |
| Oropharynx | 1.16 | ( | 0.58 | – | 2.32 | ) | 0.667 | 1.11 | ( | 0.48 | – | 2.57 | ) | 0.810 |
| Nasopharynx | 1.74 | ( | 0.70 | – | 4.31 | ) | 0.232 | 2.12 | ( | 0.76 | – | 5.91 | ) | 0.152 |
| Hypopharynx | 1.95 | ( | 0.45 | – | 8.41 | ) | 0.369 | 2.34 | ( | 0.53 | – | 10.40 | ) | 0.262 |
| Oral cavity | 11.17 | ( | 2.49 | – | 50.23 | ) | 0.002 | 13.90 | ( | 2.98 | – | 64.98 | ) | 0.001 |
| Other | 1.69 | ( | 0.39 | – | 7.24 | ) | 0.481 | 1.07 | ( | 0.14 | – | 8.19 | ) | 0.946 |
| Extent of disease | | | | | | | | | | | | |
| Localized | Ref | | | | | | Ref | | | | | |
| Regional | 1.23 | ( | 0.74 | – | 2.06 | ) | 0.421 | 1.50 | ( | 0.77 | – | 2.91 | ) | 0.235 |
| Surgery | | | | | | | | | | | | |
| Yes vs no | 0.91 | ( | 0.59 | – | 1.43 | ) | 0.690 | 0.92 | ( | 0.54 | – | 1.57 | ) | 0.770 |

Abbreviations: HR, hazard ratio, CI, confidence interval

esophagus and found that surgery conferred a survival benefit for esophageal cancer with tumor location in the lower one-third of the esophagus or tumor length of >5 cm [11]. In contrast, Meng et al. reported a poorer prognosis in patients with SmCC of the esophagus treated with surgery and chemotherapy than in those treated with CRT [12]. Chen et al. compared primary surgery versus radiation therapy for stage I/II SmCC of the uterine cervix [13]. The locoregional failure rate was higher in the patients who underwent surgery than in those who underwent radiation therapy. Therefore, the role of surgery in the treatment of SmCC remains controversial.

In our cohort, surgery was performed in 41% of the patients with HNSmCC. However, determining whether surgery should be performed is difficult. Extensive head and neck surgery can cause disfigurement, impaired speech, and aspiration. However, surgery can improve locoregional control and prolong survival. Therefore, it is imperative to determine the survival benefits of surgery. Yu et al. reported no significant differences in the survival of patients with HNSmCC treated with different treatment modalities [6]. Matsuyama et al. analyzed 39 cases of HNSmCC in Japan and concluded that CRT should be recommended for patients with HNSmCC without distant metastasis [9]. They also stated that surgery in combination with chemotherapy could be an option for patients with stage I HNSmCC. However, no results supporting the use of surgery were provided. To assess the role of surgery in the multimodal treatment of HNSmCC more accurately, we conducted a propensity score-matched study. Our

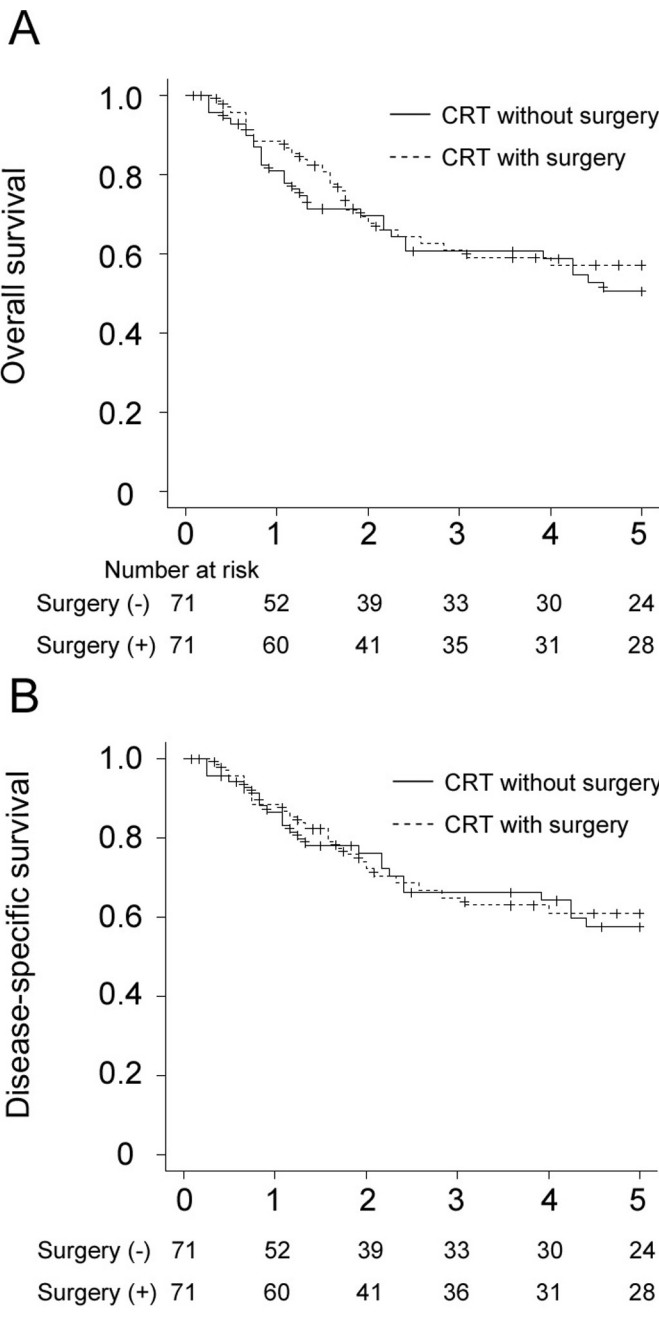

**Fig 3.** Kaplan-Meier curves for overall survival (A) and disease-specific survival (B) according to the inclusion of surgery.

results showed comparable survival rates between patients treated with CRT with and without surgery. Therefore, CRT without surgery is the standard therapy for locoregional HNSmCC.

The SEER database is an invaluable source of information on rare cancers. The SEER 18 dataset used in our study covered 27.8% of the United States population. Owing to its large sample size, we identified several patients with HNSmCC, which enabled us to perform propensity score matching. However, our SEER data analysis has several limitations. First, the SEER database lacks information on patient status, including performance status and

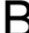

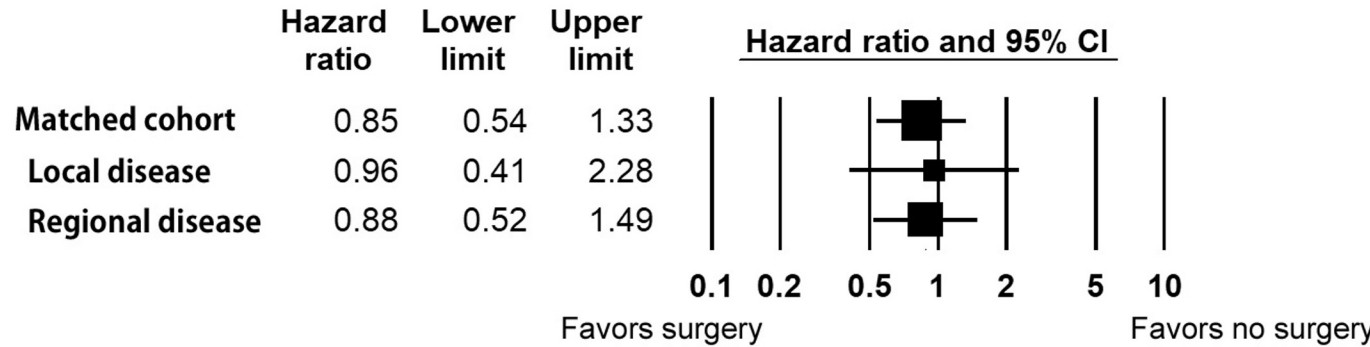

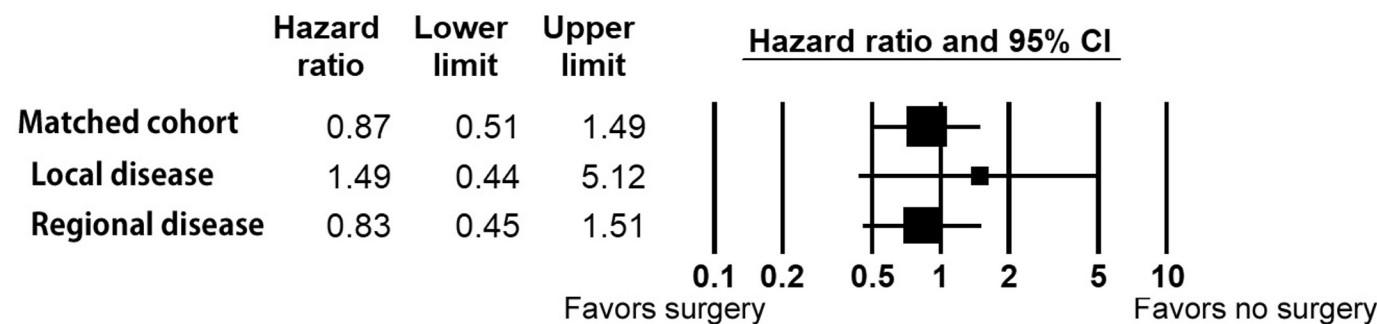

**Fig 4.** Hazard ratio for the comparison of chemoradiation therapy with and without surgery for overall survival (A) and disease-specific survival (B). Square and horizontal bars show hazard ratio and 95% confidence interval, respectively.

comorbidities. As a result, propensity score matching in our study could not fully adjust for patient background. Second, detailed information on treatment is lacking in the SEER data. SEER data does not specify the chemotherapy regimen, dose, or timing. Similarly, the radiation dose and field are not detailed. Therefore, we could not determine the recommended treatment regimen and CRT sequence. Third, the SEER database provides limited information on oncologic outcomes. Specifically, the data on recurrence and objective responses are not included. Therefore, we could not determine the response rate to CRT, progression-free survival, or cause of treatment failure.

In conclusion, we demonstrated that surgery combined with CRT did not improve the survival of patients with HNSmCC. However, our results are insufficient to guide clinicians in HNSmCC treatment. To date, HNSmCC is generally treated according to the guidelines for lung SmCC. The combination of treatment modalities should differ according to the primary site. Given the rarity of HNSmCC, an international, multi-institutional study is required.

## Supporting information

**S1 Fig. Kaplan-Meier curves stratified by clinical variables.** (A) male, (B) female, (C) age $\geq$ 65 years, (D) age < 65 years, (E) localized disease, (F) regional disease.
(TIF)

**S1 File. Patient characteristics before matching.**
(CSV)

**S2 File. Patient characteristics after matching.**
(TXT)

## Author Contributions

**Conceptualization:** Yukinori Takenaka, Hidenori Inohara.

**Data curation:** Kiyohito Hosokawa, Yukinori Takenaka, Takashi Sato.

**Formal analysis:** Kiyohito Hosokawa, Yukinori Takenaka, Takahito Fukusumi.

**Funding acquisition:** Koji Kitamura.

**Investigation:** Yukinori Takenaka, Hirotaka Eguchi, Koji Kitamura.

**Methodology:** Yukinori Takenaka, Takeshi Tsuda.

**Resources:** Takashi Sato.

**Software:** Takeshi Tsuda, Takahito Fukusumi.

**Supervision:** Motoyuki Suzuki, Hidenori Inohara.

**Validation:** Masami Suzuki, Takahito Fukusumi.

**Visualization:** Takeshi Tsuda.

**Writing – original draft:** Kiyohito Hosokawa.

**Writing – review & editing:** Yukinori Takenaka, Motoyuki Suzuki.

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
