## [Decision Letter · Decision Letter 0]

15 Aug 2024

PONE-D-24-29008Small-cell carcinoma in the head and neck region: a propensity score-matched analysis of the effect of surgeryPLOS ONE

Dear Dr. Takenaka,

Thank you for submitting your manuscript to PLOS ONE. After careful consideration, we feel that it has merit but does not fully meet PLOS ONE’s publication criteria as it currently stands. Therefore, we invite you to submit a revised version of the manuscript that addresses the points raised during the review process.

We look forward to receiving your revised manuscript.

Kind regards,

Jian Hao

Academic Editor

PLOS ONE

Journal Requirements:

This work was supported by JSPS KAKENHI Grant Number JP24K12647. The funder had no role in study design, data collection, analysis, decision to publish, or preparation of the manuscript.

4. We are unable to open your Supporting Information file "locoregional SRC RC.jmp". Please kindly revise as necessary and re-upload.

Reviewers' comments:

Reviewer's Responses to Questions

**Comments to the Author**

1. Is the manuscript technically sound, and do the data support the conclusions?

Reviewer #1: Partly

Reviewer #2: Yes

2. Has the statistical analysis been performed appropriately and rigorously? 

Reviewer #1: No

Reviewer #2: Yes

3. Have the authors made all data underlying the findings in their manuscript fully available?

Reviewer #1: Yes

Reviewer #2: Yes

4. Is the manuscript presented in an intelligible fashion and written in standard English?

Reviewer #1: Yes

Reviewer #2: Yes

5. Review Comments to the Author

**Reviewer #1: **This manuscript primarily discusses the benefits of surgery for head and neck small cell carcinoma compared to chemoradiation therapy. The use of propensity score matching to adjust for age-related effects in the surgery group is a strong aspect that refines the results. However, there are a few limitations that need to be addressed:

1. Terminology Consistency: Please review lines 57-58 to ensure consistent use of terminology. Specifically, please confirm whether the T, N, M stratification should be referred to as T, N, M stages in accordance with the reference publication by Yu et al.

2. Patient Enrollment: Please include a flow diagram to illustrate the patient selection process.

3. Propensity Score Matching: The manuscript mentions the use of 1:1 nearest matching to adjust for discrepancies in age and primary sites. However, it should be noted that nearly 40% of the sample data were not matched, indicating the current matching did not fully account for primary sites. So please consider adjusting the parameters in the propensity score matching to make better use of the data. For instance, switching to partial exact matching for primary sites, using Mahalanobis distance in MatchIt, etc. Additionally, there is no evaluation provided for the matched pairs. Please include assessment metrics, such as the distribution of propensity scores before and after matching, to demonstrate the effectiveness of the matching process.

4. Subgroup Analysis: It appears that sex and cancer stages may influence the effect of surgery, as indicated by the univariate factors and in the discussion section. Please consider conducting Kaplan-Meier (KM) curves for these subgroups to verify these effects.

**Reviewer #2:** In this study, SEER data were used to investigate overall survival and disease-specific survival in patients with locoregional small cell carcinoma of the head and neck treated with chemoradiotherapy (CRT) or CRT plus surgery. Although the study targeted a rare histological type, a sufficient number of propensity score-matched cases (71 in each group) were used for comparison.

It seems that in many cases of CRT plus surgery, small cell carcinoma was diagnosed through surgery and then CRT was added. For clinical decision-making, the prognosis of cases treated with surgery alone and surgery plus chemotherapy is also important. However, when there are multiple divergent treatments, it is difficult to compare them.　 Although this study has the limitation that it was not prospective, it clearly demonstrated that there is no benefit to performing surgery before CRT in locoregional tumors diagnosed as small cell carcinoma by biopsy.

This study will be useful to many clinicians and is worthy of publication in the journal.

6. PLOS authors have the option to publish the peer review history of their article (what does this mean?). If published, this will include your full peer review and any attached files.

Reviewer #1: No

Reviewer #2: No

---

## [Author Response · Author response to Decision Letter 0]

22 Sep 2024

As requested, I have corrected the funding information and added the following funding statement in the cover letter.

This work was supported by a grant awarded to KK by the Japan Society for the Promotion of Science (Grant Number: JP24K12647, https://www.jsps.go.jp/).

The funder had no role in study design, data collection, analysis, decision to publish, or preparation of the manuscript.

There was no additional external funding received for this study.

---

## [Decision Letter · Decision Letter 1]

8 Oct 2024

Small-cell carcinoma in the head and neck region: a propensity score-matched analysis of the effect of surgery

PONE-D-24-29008R1

Dear Dr. Takenaka,

We’re pleased to inform you that your manuscript has been judged scientifically suitable for publication and will be formally accepted for publication once it meets all outstanding technical requirements.

Kind regards,

Jian Hao

Academic Editor

PLOS ONE

Additional Editor Comments (optional):

Reviewers' comments:

Reviewer's Responses to Questions

**Comments to the Author**

1. If the authors have adequately addressed your comments raised in a previous round of review and you feel that this manuscript is now acceptable for publication, you may indicate that here to bypass the “Comments to the Author” section, enter your conflict of interest statement in the “Confidential to Editor” section, and submit your "Accept" recommendation.

Reviewer #1: All comments have been addressed

2. Is the manuscript technically sound, and do the data support the conclusions?

Reviewer #1: Yes

3. Has the statistical analysis been performed appropriately and rigorously? 

Reviewer #1: Yes

4. Have the authors made all data underlying the findings in their manuscript fully available?

Reviewer #1: Yes

5. Is the manuscript presented in an intelligible fashion and written in standard English?

Reviewer #1: Yes

6. Review Comments to the Author

Reviewer #1: Dear Authors,

Thank you for your detailed revisions. The updated manuscript demonstrates a comprehensive response to the concerns raised in the previous review, and the incorporation of evaluation of propensity score matching and subgroup analysis has greatly strengthened the study. Overall, the revisions have addressed all major concerns, and the manuscript is now significantly improved for publication. Congratulations.

7. PLOS authors have the option to publish the peer review history of their article (what does this mean?). If published, this will include your full peer review and any attached files.

Reviewer #1: No

---

## [Editor Report · Acceptance letter]

14 Oct 2024

PONE-D-24-29008R1 

PLOS ONE

Dear Dr. Takenaka, 

I'm pleased to inform you that your manuscript has been deemed suitable for publication in PLOS ONE. Congratulations! Your manuscript is now being handed over to our production team.

Kind regards, 

on behalf of

Dr. Jian Hao 

Academic Editor

PLOS ONE